# Digital Health Literacy and Person-Centred Care: Co-Creation of a Massive Open Online Course for Women with Breast Cancer

**DOI:** 10.3390/ijerph20053922

**Published:** 2023-02-22

**Authors:** Yolanda Álvarez-Pérez, Andrea Duarte-Díaz, Ana Toledo-Chávarri, Analía Abt-Sacks, Vanesa Ramos-García, Alezandra Torres-Castaño, Amado Rivero-Santana, Lilisbeth Perestelo-Pérez

**Affiliations:** 1Canary Islands Health Research Institute Foundation (FIISC), 38109 Tenerife, Spain; 2Network for Research on Chronicity, Primary Care, and Health Promotion (RICAPPS), 38109 Tenerife, Spain; 3The Spanish Network of Agencies for Health Technology Assessment and Services of the National Health System (RedETS), 28029 Madrid, Spain; 4Evaluation Unit (SESCS), Canary Islands Health Service (SCS), 38109 Tenerife, Spain

**Keywords:** breast cancer, digital health literacy, person-centred care, health education, MOOC

## Abstract

The diagnosis of breast cancer (BC) can make the affected person vulnerable to suffering the possible consequences of the use of low-quality health information. Massive open online courses (MOOCs) may be a useful and efficient resource to improve digital health literacy and person-centred care in this population. The aim of this study is to co-create a MOOC for women with BC, using a modified design approach based on patients’ experience. Co-creation was divided into three sequential phases: exploratory, development and evaluation. Seventeen women in any stage of BC and two healthcare professionals participated. In the exploratory phase, a patient journey map was carried out and empowerment needs related to emotional management strategies and self-care guidelines were identified, as well as information needs related to understanding medical terminology. In the development phase, participants designed the structure and contents of the MOOC through a Moodle platform. A MOOC with five units was developed. In the evaluation phase, participants strongly agreed that their participation was useful for the MOOC’s development and participating in the co-creation process made the content more relevant to them (experience in the co-creation); most of the participants positively evaluated the content or interface of the MOOC (acceptability pilot). Educational interventions designed by women with BC is a viable strategy to generate higher-quality, useful resources for this population.

## 1. Introduction

Breast cancer (BC) represents one of the most frequently diagnosed cancers in women worldwide [1]. According to the most recent data from the European Cancer Information System (ECIS), there were approximately 355,460 new cases of BC diagnosed across Europe in 2020, with 34,088 of those cases occurring in Spain [2]. However, thanks to early diagnosis and therapeutic advances, BC survival has increased in recent years [3], with a survival rate of around 85% [4]. Increasing prevention and treatment for BC have lowered mortality, but the diagnosis and treatment continue to have a significant impact in many areas of patients’ lives (physical, emotional, cognitive and social) [5]. The diagnosis of BC, which in most cases necessitates an effort to adjust and adapt to the new situation [5,6], is typically perceived as a traumatic event with a significant impact on the health-related quality of life of the women who suffer from it, making them more vulnerable to the potential consequences of using biased or low-quality health information.

Person-centred care (PCC) is defined as the provision of care that considers a patient’s clinical needs, life circumstances, and personal values and preferences [7]. A central component of PCC is to ensure quality communication between patients and healthcare professionals, with the aim of fostering the process of shared decision making (SDM) [8]. SDM-based interventions, such as patient decision aids (PtDAs), have been shown to improve patients’ knowledge about available treatments and their benefits/risks, decisional conflict and other decisional process variables [9]. There is a need to develop interventions to increase knowledge about PCC and digital health literacy (DHL) [10,11], particularly in chronic pathologies such as BC, where the impact of their diagnosis or treatment may increase the number of queries on the Internet and directly influence the understanding of health information [12]. Health literacy (HL) integrates the skills and motivation to find, understand, evaluate and use health information. As a result, HL facilitates informed decision making and improves the ability to manage and address health disparities, giving patients more autonomy and empowerment to take responsibility for their own health, as well as the health of their families and communities. In turn, low HL impacts health outcomes and health-related costs, leading to inefficient healthcare utilization and delivery [13]. DHL is an extension of HL that employs the same operational definition but in the context of information and communication technology resources. It involves both the provision of information and the degree to which information is understood. When these skills are lacking, technology solutions have the potential to either promote or hinder HL [14]. Due to the complexity of health information, it is recommended that DHL interventions be based on a design of co-creation of resources, websites and health tools through collaborative work with patients, allowing them to improve the medical care they receive [15,16,17].

Massive open online courses (MOOCs) are designed to engage a large number of participants learning remotely, offering the general population, clinical subpopulations or health professionals good quality knowledge on health issues through interactive and flexible technological resources, with little or no prior learning required [18]. To date, most MOOCs have been developed for the education of medical students and health professionals [19,20], but they have also been directed at the general population or clinical subpopulations, showing positive effects in several areas such as healthy nutrition habits [21], self-management of diabetes [22] or learning risk factors for dementia [23].

As has been observed in some projects with other populations, the development of educational interventions with a MOOC based on a co-creation design, which combines several resources in different formats and adapts to different educational, cultural levels and needs of the users, could be a strategy to face the HL, self-care and empowerment challenges for women with BC. One example is the IC-Health European project (https://cordis.europa.eu/project/id/727474/es accessed on 20 December 2022), whose results have shown good acceptance of co-created teaching resources aimed at improving the DHL of people with chronic diseases and the general population [24,25,26].

In recent years, the framework of participatory action research has been used for the development of eHealth. It is an approach that involves collaboration to develop a process through the construction of knowledge and social change in a community following a cyclical approach and involving stakeholders as co-investigators in the process [27,28]. As occurs in other participatory processes, the co-design of health interventions contributes to improving the services offered, to the extent that they are adjusted to the needs and priorities of its participants while incorporating their own skills [29,30,31].

In general, digital interventions, such as MOOCs, have the potential to improve the quality of life and outcomes for women with BC by providing access to information from anywhere at any time, thereby increasing accessibility and flexibility, as well as support to complement traditional medical treatments. Therefore, the aim of this study is to co-create a MOOC of PCC and DHL for and with women with BC.

## 2. Materials and Methods

### 2.1. Design

The MOOC was co-created using a modified experience-based design approach [32]. The co-creation process was divided into three sequential phases: (a) exploratory phase, (b) development phase and (c) evaluation phase (see Section 2.3).

### 2.2. Participants and Recruitment

Adult women (≥18 years) in any of their cancer stages and BC survivors (regardless of DHL level and knowledge about PCC), their families/carers and any healthcare professionals involved in the management of BC (oncologists, gynaecologists, nurses, psycho-oncologists, etc.) were invited to participate voluntarily in the MOOC co-creation process. A theoretical sample optimized the maximum variability of sociodemographic and clinical profiles (age, educational level, time since diagnosis and active treatment) of women with BC. The recruitment was carried out via snowball sampling [33] through healthcare professionals and expert patients (BC survivors) between May and June 2020. Participants signed an informed consent declaration. 

### 2.3. Procedure

The co-creation process was carried out in three online sessions of 120 min each (via the Zoom platform due to the COVID-19 pandemic and delivered by members of the research team) between June 2020 and March 2021 and was supported by a Moodle platform.

The first session (exploratory phase) was held in June 2020 and consisted of (i) a brief presentation of the participants; (ii) identifying the different diagnosis, treatment and long-term follow-up paths for BC represented through a patient journey map (PJM)—a scheme that aims to reflect the care pathway followed by a person [5,32]—based on their experiences, emotions, feelings and thoughts; (iii) exploring their empowerment and information needs in each phase of the disease; (iv) and exploring patients’ information needs and experiences on patient empowerment and SDM. Health professionals did not participate in the development of the PJM; they offered advice and their experiences on the most frequent concerns found in clinical practice with these patients, according to the phase of the disease.

In the second session (development phase), held in July 2020, the participants reviewed the PJM and designed the structure and proposed the contents of the MOOC (self-care, myths related to BC, strategies to improve DHL, etc.) based on the empowerment and information needs identified in the first session and their previous experiences managing BC information online. At the end of this session, participants were encouraged to continue the process of co-creation online between July and December 2020 through a Moodle platform where the participants were registered and which they accessed with an individual username and password (assessment phase). The research team developed and shared some content proposals weekly for the different units of the MOOC, and participants were asked to provide feedback and/or new content proposals (see Section 3.3). Initially, the content of the units was presented in infographic format (see Appendix A) and was mainly related to PCC, self-care and DHL applied to BC. Once all the suggestions for improvement provided by the participants on the content were compiled, a graphic designer developed videos and edited the infographics to provide them with interactivity and visually improve their appearance. Updated contents were shared again with the participants in March 2021. Through questionnaires on the Moodle platform (see Section 2.4.2), they could give feedback on the definitive contents of the MOOC (see Section 2.4.1).

A third session (evaluation phase) was held in March 2021 to offer final feedback about the content and interface of the MOOC (acceptability pilot) and to evaluate the experience in the co-creation of the MOOC by means of specific questionnaires (see Section 2.4). Four gift cards were raffled off as a token of appreciation during this last meeting.

### 2.4. Measures

#### 2.4.1. Experience in the Co-Creation Process

A 13-item questionnaire was specifically developed to explore patients’ and healthcare professionals’ experience in the co-creation process. The first 6 items were measured using a 5-point Likert scale (from “strongly disagree” to “strongly agree”), addressing satisfaction with communication, objective adequacy, usefulness of patient involvement in the co-creation process, importance of co-creation to design relevant content for patients, self-perception of increased knowledge and feeling of being part of the team project. The following 4 items were also assessed on a 5-point Likert scale (from “insufficient” to “excellent”) and were related to participants’ opinions on the quality and clarity of the co-creation sessions, the methodology employed, the interactions between participants and the researchers’ implication. The last 3 items were open-ended questions about what participants liked the most and the least about the MOOC co-creation process, which aspects they found most useful and which aspects could be improved in the co-creation process (see Section 3.4).

#### 2.4.2. Acceptability Pilot of the MOOC

The MOOC’s acceptability was evaluated using a specific scale created in the context of the project following the technology acceptance model’s (TAM) methodology [34] and based on previous related studies [35,36]. This scale assessed factors such as ease of navigation, clarity of objectives and language, appropriateness of learning activities and quizzes, and other characteristics of the MOOC. The acceptability questionnaire, answered by both patients and healthcare professionals, included 18 items: the first 15 were rated on a 5-point Likert scale, and the last 3 items were open-ended questions about strengths and weaknesses, improvement suggestions and the main points learned throughout the MOOC (see Section 3.5). 

### 2.5. Analysis

The PJM and MOOC content were progressively developed in conjunction with participants. A draft was created with the information obtained from the online co-creation sessions. The different sections of the PJM summarize the experiences of participants with BC or survivors. The research group reviewed the contributions of the participants and proposed a draft version based on a PCC framework. Subsequently, this version of PJM and MOOC content was reviewed by all participants through an iterative process until consensus was reached.

For the experience in the co-creation process and the acceptability pilot of the MOOC measures, means and standard deviations (SD) were calculated for all items assessed, and we also analysed the response distribution for each item.

## 3. Results

Twenty-eight participants from Tenerife and Gran Canaria (Canary Islands, Spain) were contacted between May and June 2020, of whom 19 participated in the co-creation process: 17 patients (Table 1) and two healthcare professionals (nurses from gynaecology and breast pathology units; mean age 40 (1.41) years and with more than 10 years of professional experience).

### 3.1. Patient Journey Map

Points of contact, experience with healthcare received, emotions, feelings and thoughts, diagnostic and therapeutic treatments, and perception about own participation in shared decision making for the three stages of the trajectory of care of BC (early detection and diagnosis, treatment and long-term follow-up) were collected on the co-designed PJM (Figure 1). 

#### 3.1.1. Early Detection and Diagnosis Stage

Most of the participants received their diagnosis during routine controls (specialized care) or as a result of the presence of symptoms (primary care), and the main emotions that emerged during this time were shock, anxiety, uncertainty and worry about the future. The main diagnostic techniques that the participants underwent were physical examination (palpation), imaging tests (mammography and ultrasound) and biopsy. The experiences collected about the healthcare received in this stage were related to the perception of professionalism, friendliness, a predisposition to resolve doubts and the transmission of calm and encouragement from the healthcare professionals who attended to them. However, the participants expressed that there were other drawbacks in the medical care received at this time related to the challenges of early detection and the complexity of some administrative processes (e.g., medical appointments). Some participants expressed that they would have liked more advice from medical staff. Other participants expressed that they felt involved in the decision-making process in this phase, and this helped them accept the disease and have trust in the therapeutic approach to be used.

#### 3.1.2. Treatment Stage

Participants identified the involvement of other healthcare professionals (e.g., oncology, gynaecology, surgery and rehabilitation, among others). While uncertainty remained the predominant emotion in this stage, other emotions started to emerge as well, including concern for appearance and shock by the physical changes that were occurring as a result of the therapeutic techniques used in this stage (e.g., chemotherapy, radiotherapy, surgery, etc.). In general, the participants experienced empathetic care and a certain psychological accompaniment by the healthcare professionals who assisted them. The participants felt more involved in the decision-making process in the gynaecology units than in the oncology units. They all concurred that the experience of informed participation in their treatment process was positive.

#### 3.1.3. Long-Term Follow-Up Stage

The main experience was less follow-up by healthcare professionals, giving rise to feelings of helplessness or loneliness and uncertainty about self-care. Other concerns, such as going back to work or looking for a new job more adapted to their health needs, were shared among the participants. The treatments at this stage focused on breast reconstruction surgery and medication. All the participants said they had received limited information on self-care, medical care to follow from this stage and possible new treatments required by healthcare professionals. However, they commented that at this stage they felt empowered to choose the aspects of their health in which they wanted to be involved, leading them to request personalized attention and to ask questions in order to be more involved in the decision-making process.

#### 3.1.4. Recommendations of the Participants for Other Women with BC or Survivors

Additionally, and at their own initiative, the participants in the co-creation sessions provided a series of recommendations or tips for other women diagnosed with BC and suggested their inclusion in the MOOC as another resource. These tips were about family, social, work and empowerment areas and specifically for each of the stages worked (Figure 2).

### 3.2. Empowerment and Information Needs

Figure 3 shows the empowerment and information needs identified in each phase. The main empowerment needs identified were related to strategies for emotional management and guidelines for self-care throughout the process from diagnosis until long-term follow-up. The main information needs were related to the lack of understanding of the meanings of biomarkers, parameters and acronyms found in reports, as well as medical jargon, treatment options and the likelihood of cancer recurrence. The need to have guidelines for accessing information and support resources available online, including association websites and online experiences of other women with BC, was highlighted.

### 3.3. MOOC Content Development

Between July and October 2020, a weekly activity was published on the Moodle platform to carry out the process of co-design of the MOOC content. Table 2 shows the themes of these activities. Finally, the MOOC was composed of five units: (i) BC (definition, types and stages, diagnostic process, treatments, myths, etc.), (ii) PCC (definition, implementation strategies, tips for preparing consultations with the healthcare professional, etc.); (iii) DHL (definition, guidelines to improve each skill, etc.), (iv) self-care (management of physical side effects, emotional management, etc.) and (v) experiences and advice from patients in different areas (healthcare, family, social and work area) and moments of the disease (diagnosis, treatment and long-term follow-up).

### 3.4. Experience in the Co-Creation Process

Data was available for seventeen participants (89.47%) (Table 3). All of them strongly agreed or agreed that the general objectives of the project were adequate (item 2) and that the participation of women who have or have had BC is useful for the development of a MOOC on this content (item 3). More than 88% of the participants strongly agreed or agreed that being part of the MOOC co-creation process made the content more relevant to them (item 4) and rated the quality of the activities carried out in the co-creation process (item 7) and the methodology applied (item 8) as very good or excellent. Regarding open questions, participants appreciated the way their experiences were incorporated into the MOOC and how they felt part of something meaningful, sharing experiences with other women in similar situations (item 11). In order to fully engage in the co-creation process, participants expressed that they would have liked to attend a face-to-face session. Additionally, some participants found it challenging to devote more time to the MOOC due to personal issues (item 12). See Appendix A to consult illustrative quotes from participants’ responses to open questions. 

### 3.5. Acceptability Pilot of the MOOC

Data was available for seven participants (36.84%) (Table 4). Combining the “totally agree” and “agree” categories, most of the participants positively evaluated the acceptability of the MOOC in terms of language, content, relevance, proposed activities and suitability of the MOOC objectives. Regarding open questions, most participants emphasized the usefulness of the MOOC’s content (especially related to SDM) and the way it is presented (through infographics and other audio-visual materials) as strengths. Nevertheless, one participant pointed out some navigation difficulties, while another emphasized the lengthy process (item 16). When it was possible, improvements suggested by participants were implemented, such as adding an initial summary of the MOOC’s content (item 17). All the contents were mentioned as important topics learned after completing the MOOC (item 18). See Appendix A to consult illustrative quotes from participants’ responses to open questions.

## 4. Discussion

This study presents the development of a MOOC aimed to improve the DHL of women with BC. We used a co-creation approach involving 17 patients and survivors and two nurses. In order to inform the content of the MOOC, we explored participants’ perceptions of the extent they were involved in the decision-making process, as well as their feelings, emotions and information needs throughout the therapeutic process. Most participants indicated that the MOOC co-creation experience was positive and made them feel involved in the project, and they positively valued the final product. Similar results were obtained by our team with other MOOCs developed for pregnant and lactating women [26] and people with type-2 diabetes [25], including larger samples than the one used in this study. In these two studies, participants’ self-perceived DHL significantly improved after completing the MOOC development compared to baseline. Future work is warranted to evaluate the effectiveness of this MOOC at improving BC patients’ actual DHL (not only self-perceived), objective knowledge of the disease and treatments, and their involvement in treatment decisions.

Women in this study pointed out information needs concerning different stages of the cancer, from diagnosis to long-term follow-up, as shown in previous studies [37,38]. Increasingly, these patients want to be involved in the decisions related to their health, and some studies have focused on involving the patient experience to improve the healthcare they receive [31,39]. As a result of an exchange of information and values between patients and healthcare providers, SDM engages patients as partners in their own care and optimizes the decision-making process [40]. To support SDM and the use of PtDA in the practice, it is important that patients also have a certain level of HL to increase patient empowerment and allow them to adopt a more participatory role in their healthcare [41]. Online interventions that provide information and support to women with BC appear to cushion the uncertainty they experience at different stages of the disease, and MOOCs can be an effective educational resource for meeting these unmet needs and promoting both DHL and SDM processes [24,39].

The PJM considered the evolving requirements for empowerment during the stages of diagnosis, treatment and long-term follow-up. Knowledge of the patients’ experiences, through a PJM, facilitates the identification of key moments in which to provide more precise information [5]. As we have seen in the results of this study, depending on the individual experiences of each woman, the care received during various BC periods could be perceived as more or less satisfactory. Based on our results, women with BC positively valued the experience of participating in the co-creation process of the MOOC, which made the content more relevant to them. This result aligns with previous evidence suggesting that a user-centred design process involves the participation of groups of users throughout the entire development cycle, during which they describe the context in which the generated resources will be used, their needs as users, and take part in user tests [42,43]. These are all contributions for designing and building health information technology through iterations [44].

This intervention represents an opportunity to reach a larger population that, due to health, availability and/or travel circumstances, may find it impossible to attend another type of face-to-face training on this subject. Technology provides great options for enhancing patient care; however, disparities in access and DHL continue to negatively impact vulnerable populations because of potential barriers in the digital sphere for those with low HL [11]. This problem can be especially aggravated as more information is provided online and healthcare professionals must be involved in the development of these skills in their patients with BC, but they also require support and a strategy at the institutional level. Therefore, healthcare organizations must prioritize achieving accessibility for all patients when designing eHealth services [11].

In this regard, the integration of educational materials designed by a representative sample of the target population to which they are addressed makes this proposal an opportunity to contribute to obtaining relevant health results for both affected patients and their healthcare professionals and, ultimately, decision-makers with financial capacity. From a managerial perspective, healthcare organizations should reframe their strategies, procedures and approaches, embracing a patient-centred perspective to become health literate [45]. From a policy perspective, it suggests that individual HL and organizational HL should be handled as two complementary tools to empower people and to engage them in self-care and health policy making [45].

The main strength of this project is having involved the intended audience in the creation of MOOCs, which enhances the significance of the material covered and how it is delivered. This is important because they have valuable insights and perspectives on the subject matter and can provide feedback on the relevance and effectiveness of the content and its delivery. This can lead to the creation of more engaging and effective MOOCs that better meet the needs and expectations of the intended audience. Nonetheless, there are several limitations to the study. Initially, it had been proposed that the co-creation process be based on face-to-face sessions with the participants followed by some online sessions through the Moodle platform. However, due to the COVID-19 pandemic, face-to-face sessions were replaced by online sessions carried out through the Zoom platform. This fact made the co-creation process last a few weeks longer than expected by adapting the work rhythms to the availability and web resources of the participants. However, the online sessions had several advantages: participants did not have to travel, the meetings were easier to organize and fewer financial resources were needed to support the development of the sessions. Another limitation is that, although all professionals related to BC were invited to participate, only two nurses did so. Perhaps the participation of other professionals involved in the process (e.g., gynaecologists and oncologists), as well as family members and/or caregivers, could have been beneficial for the generation of more useful resources. Even though women of all educational levels participated, the majority had higher education, so there was not much variability in this regard and lower educational levels may have been under-represented. In addition, there is a need for independent evaluation of acceptability to confirm the results obtained. Likewise, it is necessary to carry out an evaluation of the effectiveness of the MOOC with an independent sample that allows us to know if there really is an improvement in the levels of DHL and a change in knowledge in all the areas that are included in the different modules of the MOOC (BC, PCC, DHL, etc.).

## 5. Conclusions

The work carried out in this project is an example of how the development of educational interventions in MOOC format directed and designed by women with BC, with resources in different formats adapted to different educational/cultural levels and needs of the users, seems to be a viable strategy to generate higher-quality and useful resources for this population. The co-creation methodology and this type of resource aim to address the literacy and empowerment challenges of women with BC.

## Figures and Tables

**Figure 1 ijerph-20-03922-f001:**
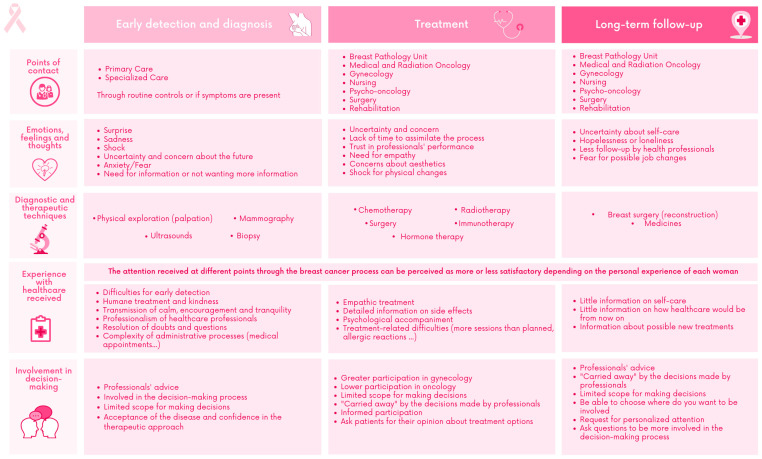
Patient Journey Map.

**Figure 2 ijerph-20-03922-f002:**
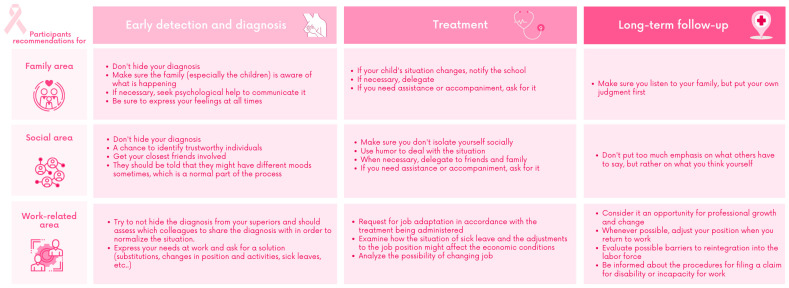
Participant recommendations for other patients.

**Figure 3 ijerph-20-03922-f003:**
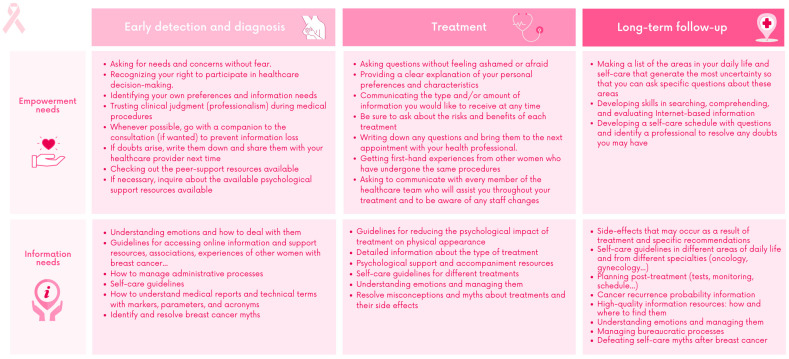
Empowerment and information needs.

**Table 1 ijerph-20-03922-t001:** Characteristics of the participants (patients).

	Participants with BC or Survivors (*n* = 17)
Age (years) (mean, sd)	47.88 (6.23)
Education level (*n*, %)	
Primary school	1 (5.88)
High school	2 (11.76)
Medium/high technical education	6 (35.29)
University degree	8 (47.07)
Employment status (*n*, %)	
Employed	6 (35.29)
Unemployed	3 (17.65)
Sick leave	8 (47.06)
Civil status (*n*, %)	
Married/living with partner	11 (64.71)
Separated or divorced	3 (17.65)
Single	3 (17.65)
Breast cancer diagnosis date (*n*, %)	
≥5 years ago	11 (64.71)
<5 years ago	6 (35.29)
Currently undergoing treatment (*n*, %)	
Yes	9 (52.94)
Hormone treatment (*n*, *%*)	5 (55.56)
Chemotherapy/radiotherapy (*n, %*)	2 (22.22)
Plastic surgery (breast reconstruction) (*n*, *%*)	2 (22.28)
No	8 (47.06)
Previous knowledge about PCC (*n*, %)	
Yes, but superficial, not deep	3 (17.65)
Yes, with previous education	1 (5.88)
No	13 (76.47)

**Table 2 ijerph-20-03922-t002:** Co-creation activities online in Moodle platform (development phase).

1. Welcome to the community of practice; Platform user manual; Presentation in the participant forum and personalize user profile; Review of the patient journey map developed in the co-creation sessions; Select images to include in MOOC contents;
2. Person-centred care module development; Review of information on person-centred care, shared decision making and PtDAs; Facilitate access to some existing PtDAs on BC;
3. Digital health literacy module development; Review of information on DHL and skills (search, understand, assessment and apply);
4. Breast cancer module development; Review of information on BC (types, treatments, symptoms, side effects, etc.); Review of information on self-care and practical advice based on the personal experiences of the participants; Review of information on common myths related to BC;
5. Propose additional materials;
6. Development of assessment questions for the MOOC; Select among possible assessment questions for each MOOC module and/or suggest alternative questions;
7. Select MOOC title; Propose a title for the MOOC; Voting among the title proposals; Selection of the most voted title;
8. Evaluation; Experience in the co-creation process assessment; Pilot evaluation of the acceptability of the MOOC on the Moodle platform; Self-perceived health literacy assessment.

**Table 3 ijerph-20-03922-t003:** Experience in the co-creation process.

Questions	Strongly Agree *n* (%)	Agree *n* (%)	Not Sure *n* (%)	Disagree *n* (%)	Strongly Disagree *n* (%)	Mean ^a^ (sd)
1. I am satisfied with the communication maintained with the researchers at the different moments of the project (in the online sessions, by WhatsApp, by the platform, by email, etc.)	14 (82.35)	2 (11.77)	1 (5.88)			4.76 (0.56)
2. The general objectives of the project are adequate	12 (70.59)	5 (29.41)				4.71 (0.47)
3. I consider that the participation of women who have or have had breast cancer is useful for the development of a MOOC on this content	16 (94.12)	1 (5.88)				4.94 (0.24)
4. Being part of the process of co-creating the MOOC made the content more relevant to me	11 (64.71)	4 (23.53)	1 (5.88)		1 (5.88)	4.41 (1.06)
5. Participating in the different online sessions has increased my knowledge about digital health literacy and this has helped me increase my ability to take control of my health	11 (64.71)	4 (23.53)	2 (11.77)			4.53 (0.72)
6. The process of co-creating the MOOC made me feel part of the project	8 (47.06)	6 (35.29)	2 (11.77)		1 (5.88)	4.18 (1.07)
	**Excellent** ***n* (%)**	**Very good** ***n* (%)**	**Good** ***n* (%)**	**Low** ***n* (%)**	**Insufficient** ***n* (%)**	**Mean ^a^ (sd)**
7. In general, how would you rate the quality of the activities carried out in the online sessions, the documents and visual resources that were used and the clarity of the presentation and the dynamics carried out?	8 (47.06)	7 (41.18)	2 (11.77)			4.35 (0.7)
8. How would you rate the online methodology that has been carried out for the development of the MOOC materials?	8 (47.06)	7 (41.18)	1 (5.88)	1 (5.88)		4.29 (0.85)
*9. How would you rate the level of interaction and participation of the rest of the participants in the development of the MOOC?*	2 (11.77)	9 (52.94)	6 (35.29)			3.76 (0.66)
*10. How would you rate the involvement of the research team in the development of the MOOC?*	14 (82.35)	3 (17.65)				4.82 (0.39)
11. Open question: What did you like most about the process of joint creation of the Online Course? What aspects do you consider most useful?						
12. Open question: What did you like least about the process of joint creation of the Online Course? What aspects do you think could be improved?						
13. Open question: If you have any further comments regarding your participation in this process, please include them						

^a^ Higher scores indicate more positive rating (range 1–5 for all items).

**Table 4 ijerph-20-03922-t004:** Acceptability of the MOOC.

Questions	Totally Agree *n* (%)	Agree *n* (%)	Not Sure *n* (%)	Disagree *n* (%)	Totally Disagree *n* (%)	Mean ^a^ (sd)
1. The language on the MOOC was easy to understand	5 (71.43)		2 (28.57)			4.43 (0.98)
2. The MOOC is easy to navigate, and the information was clearly organized	4 (57.14)	2 (28.57)	1 (14.29)			4.43 (0.79)
3. The information presented was organized in a coherent and clear way	4 (57.14)	2 (28.57)	1 (14.29)			4.43 (0.79)
5. The objectives of the MOOC were clear	5 (71.43)	1 (14.29)	1 (14.29)			4.57 (0.79)
6. The content of the MOOC was consistent with the proposed objectives and with the people to whom the MOOC is directed	6 (85.71)	1 (14.29)				4.86 (0.38)
7. I find the MOOC interesting	5 (71.43)	1 (14.29)	1 (14.29)			4.57 (0.79)
8. This MOOC has met my expectations	5 (71.43)	1 (14.29)	1 (14.29)			4.57 (0.79)
9. The description of the content of each unit and of the activities were clear, avoiding possible errors in their interpretation	5 (71.43)	1 (14.29)	1 (14.29)			4.57 (0.79)
10. There was consistency between the name of a section or link and the content that was displayed or the site to which it was directed	5 (71.43)	1 (14.29)	1 (14.29)			4.57 (0.79)
11. The activities proposed in each unit were useful to deepen the understanding of the content worked	5 (71.43)	1 (14.29)	1 (14.29)			4.57 (0.79)
12. The final evaluation questions of each unit adequately assessed the material presented in the MOOC	3 (42.86)	2 (28.57)	2 (28.57)			4.14 (0.9)
14. This MOOC has been able to improve my ability to self-care in my own health by learning to access relevant and reliable information that I can obtain from the Internet	4 (57.14)	2 (28.57)	1 (14.29)			4.43 (0.79)
15. I would recommend this MOOC to other people	6 (85.71)	1 (14.29)				4.86 (0.38)
	**Very high quality** ***n* (%)**	**High quality** ***n* (%)**	**Not Sure** ***n* (%)**	**Low quality** ***n* (%)**	**Very low quality** ***n* (%)**	**Mean ^a^ (sd)**
4. The quality of the general design of the course and its contents seemed to me	4 (57.14)	2 (28.57)	1 (14.29)			4.43 (0.79)
	**Appropriate** ***n* (%)**	**Insufficient, requires spending very little time** ***n* (%)**		**Excessive, requires spending too much time** ***n* (%)**		**Mean ^a^ (sd)**
13. The amount of time needed to spend to view all the MOOC content is	6 (85.71)			1 (14.29)		2.86 (0.38)
16. Open question: Please provide a short summary of the strengths and weaknesses of the MOOC						
17. Open question: Please provide brief suggestions on how to improve the MOOC						
18. Open question: What are the main points that you have learned through this MOOC?						

^a^ Higher scores indicate more positive rating (range 1–5 for items 1–3 and 5–15; range 1–3 for item 4).

## Data Availability

The data presented in this study are available in Appendix A.

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
