# Peer review of "Digital Health Literacy and Person-Centred Care: Co-Creation of a Massive Open Online Course for Women with Breast Cancer"

_ijerph, 2023, doi:10.3390/ijerph20053922_

Round 1
Reviewer 1 Report
-Line153 : Was a study conducted on the scale to determine its validity?(The acceptability of the MOOC was evaluated through a specific instrument created Line 165
in the context of the project following the Technology Acceptance Model’s (TAM) methodology
-line 180: Was the Cronbach alpha test performed and what was the result?
-Line 196: What is the psychological theoretical basis on which the table is classified? Figure 1. Patient Journey Map
-Who was the one who determined the topics of dissemination of information through the platform and whether the topics were among the needs of the participants
-What are the determinants, strengths and weaknesses of the research?
Author Response
Dear editor, thank you very much for giving us the opportunity to modify and improve the article based on the valuable comments of the reviewers. We have made changes to the manuscript according to your suggestions. Next, we detail our response to each of your comments point by point.
Point 1: Line 153: Was a study conducted on the scale to determine its validity? (The acceptability of the MOOC was evaluated through a specific instrument created in the context of the project following the Technology Acceptance Model’s (TAM) methodology).
Response 1: The acceptability scale is not a psychometric scale, since it does not pretend to assess a latent psychological construct by means of correlated indicators (items), and there is not a total score. Items assess how different dimensions of the MOOC are evaluated by users, and their empirical correlation is not relevant for our purposes.
For the development of this scale we based on the Technology Acceptance Model (TAM) methodology, used in other similar contexts about acceptance of information technology, as well as on previous related studies. We have included another reference and expanded the wording of section “2.4.2. Acceptability pilot of the MOOC”:
2.4.2. Acceptability pilot of the MOOC
The MOOC's acceptability was evaluated using a specific scale created in the context of the project following the Technology Acceptance Model’s (TAM) methodology [34] and based on previous related studies [35,36]. This scale assessed factors such as ease of navigation, clarity of objectives and language, appropriateness of learning activities and quizzes, and other characteristics of the MOOC. The acceptability questionnaire, answered by both patients and healthcare professionals, included 18 items: the first 15 were rated on a 5-point Likert scale and the last 3 items with open-ended questions about strengths and weaknesses, improvement suggestions and the main points learned throughout the MOOC (see 3.5. Acceptability pilot of the MOOC).
References added:
- García Toribio, G.; Polvo Saldaña, Y.; Hernández Mora, J.J.; Sánchez Hernández, M.J.; Nava Bautista, H.; Collazos Ordóñez, C.A.; Hurtado Alegría, J.A. Medición de la usabilidad del diseño de interfaz de usuario con el método de evaluación heurística: dos casos de estudio. Rev. Colomb. Comput. 2019, 20, 23–40, doi:10.29375/25392115.3605.
- Yılmaz, N.G.; Sungur, H.; van Weert, J.C.M.; van den Muijsenbergh, M.E.T.C.; Schouten, B.C. Enhancing patient participation of older migrant cancer patients: needs, barriers, and eHealth. Ethn. Health 2022, 27, 1123–1146, doi:10.1080/13557858.2020.1857338
Point 2: Line 180: Was the Cronbach alpha test performed and what was the result?
Response 2: As commented above for the acceptability scale, we did not calculate a total score for the experience measure since we were interested in the individual items, and therefore the Cronbach alpha was not calculated.
Point 3: Line 196: What is the psychological theoretical basis on which the table is classified? Figure 1. Patient Journey Map.
Response 3: The theoretical background of Patient Journey Maps (PJM) is based on the study of patients' experiences which considers care as relational and patient experiences across their journey of illness a source of specific knowledge. Experiential patient knowledge differs from knowledge gained through biomedical, research or clinical practice and is essential to improve or redesign health care services (Ziebland, 2013, Bates 2006). PJM visually describe patients' experience of health services and are used for their redesign (MacCarthy 2016).
- Ziebland, S., Coulter, A., Calabrese, J.D. y Locock L. (Eds.) Understanding and Using Health Experiences. Improving patient care. Oxford University Press: Londres.
- Bate P, Robert G. Experience-based design: from redesigning the system around the patient to co-designing services with the patient. Qual. Saf. Health Care. 2006 Oct;15(5):307–10.
- McCarthy S, O’Raghallaigh P, Woodworth S, Lim YL, Kenny LC, Adam F. An integrated patient journey mapping tool for embedding quality in healthcare service reform. J. Decis. Syst. 2016;25:354–68.
Point 4: Who was the one who determined the topics of dissemination of information through the platform and whether the topics were among the needs of the participants.
Response 4: The research team developed and shared weekly some content proposals for the different units of the MOOC. These proposals were based on the contents of the MOOC that the participants proposed in the first session (regarding self-care, myths related to BC, strategies to improve DHL, etc.), as well as their empowerment and information needs.
In section “2.3 Procedure” we explain that: “In the second session (development phase), held in July 2020, the participants designed the structure and proposed the contents of the MOOC (self-care, myths related to BC, strategies to improve DHL, etc.) based on the empowerment and information needs identified in the first session and their previous experiences managing BC information online.” (Lines 128-132), and subsequently, “The research team developed and shared weekly some content proposals for the different units of the MOOC and participants were asked to provide feedback and/or new content proposals (see 3.3. MOOC content development) (Lines 135-138).
Point 5: What are the determinants, strengths and weaknesses of the research?
Response 5: We have added an initial paragraph highlighting the strengths of the study in the Discussion section, which is followed by the limitations of the study. The wording is as follows:
“The main strength of this project is involving the intended audience in the creation of MOOCs, which enhances the significance of the material covered and how it is delivered. This is important because they have valuable insights and perspectives on the subject matter, and can provide feedback on the relevance and effectiveness of the content and its delivery. This can lead to the creation of more engaging and effective MOOCs that better meet the needs and expectations of the intended audience. Nonetheless, there are some limitations to the study. Initially, it had been proposed [...]”.

Reviewer 2 Report
The paper presents an open online course for digital health literacy and person-centered care. Different stages are discussed with necessary details. Some points need to be addressed before the paper is published.
The authors need to discuss the need for the system in the abstract.
Add organization of the paper at the end of the introduction section.
No information related to the existing systems is presented in the paper. There should be discussion on state-of-the-art methods and similar courses/studies with the strength and weaknesses of each. A paragraph covering the need/motivation for the course and the novelty of the proposed course needs to be included.
The last point in Table 1 is confusing. 13 patients had no previous knowledge of PCC? They are more than 18 years of age and cancer patients. They should have some knowledge (not zero).
The discussion section needs to be focused on the findings of the study that adds knowledge to the literature. The discussion is presented like the introduction section.
The English version of Figure S1 needs to be included.
Author Response
Dear editor, thank you very much for giving us the opportunity to modify and improve the article based on the valuable comments of the reviewers. We have made changes to the manuscript according to your suggestions. Next, we detail our response to each of your comments point by point.
Point 1: The paper presents an open online course for digital health literacy and person-centered care. Different stages are discussed with necessary details. Some points need to be addressed before the paper is published. The authors need to discuss the need for the system in the abstract.
Response 1: We have included a brief sentence commenting the potential usefulness of MOOCs to improve digital health literacy and person-centred care.
Point 2: No information related to the existing systems is presented in the paper. There should be discussion on state-of-the-art methods and similar courses/studies with the strength and weaknesses of each. A paragraph covering the need/motivation for the course and the novelty of the proposed course needs to be included.
Response 2: We add a paragraph in the Introduction section about the relevance of creating a MOOC aimed at women with BC.
Point 3: The last point in Table 1 is confusing. 13 patients had no previous knowledge of PCC? They are more than 18 years of age and cancer patients. They should have some knowledge (not zero).
Response 3: Table 1 provides the participant’s answer to the question “Do you know or have you heard of Person-Centered Care or any of its components (shared decision-making, empowerment,...)?”. The response options were “No” or “Yes” (if the participants responded “Yes”, other questions about how they obtained that knowledge (very superficially without going into detail or delving deeper; autodidact education; or received previous formal training) were presented). Thirteen patients responded “No” to previous knowledge about PCC.
Point 4: The discussion section needs to be focused on the findings of the study that adds knowledge to the literature. The discussion is presented like the introduction section.
Response 4: We have included a new paragraph and made other changes in the discussion section.
Point 5: The English version of Figure S1 needs to be included.
Response 5: Figure S1 has been translated into English.
